# How Do Trade-Offs and Synergies between Ecosystem Services Change in the Long Period? The Case Study of Uxin, Inner Mongolia, China

**Jing Zhang [1,2], Xueming Li [1,*], Alexander Buyantuev [3] , Tongliga Bao [4] and Xuefeng Zhang [4]**

1   School of Geography, Liaoning Normal University, Dalian 116029, China; zhangjing@dlnu.edu.cn
2   College of Environment and Resources of Dalian Minzu University, Dalian 116600, China
3   University at Albany, State University of New York, Albany, NY 12222, USA; abuyantuev@albany.edu
4   College of Resources and Environment, Baotou Teachers' College, Baotou 014030, China;
    tongliga113@126.com (T.B.); xfzhang2003@163.com (X.Z.)
*   Correspondence: lixueming999@163.com; Tel.: +86-411-8215-8258

**Abstract:** Ecosystem services management should often expect to deal with non-linearities due to trade-offs and synergies between ecosystem services (ES). Therefore, it is important to analyze long-term trends in ES development and utilization to understand their responses to climate change and intensification of human activities. In this paper, the region of Uxin in Inner Mongolia, China, was chosen as a case study area to describe the spatial distribution and trends of 5 ES indicators. Changes in relationships between ES and driving forces of dynamics of ES relationships were analyzed for the period 1979–2016 using a stepwise regression. We found that: the magnitude and directions in ES relationships changed during this extended period; those changes are influenced by climate factors, land use change, technological progress, and population growth.

**Keywords:** ecosystem services; trade-offs and synergies; driving forces; Uxin

## 1. Introduction

Ecosystem services (ES) refer to the conditions and utilities of the natural environment, as well as to the benefits people obtain from these ecosystems [1]. ES not only maintain the life cycle process, ensure biodiversity, purify the environment, and perform other functions and ecological processes humans rely on, but they also provide raw materials, food, fiber, clean water and recreation, which are necessary for human well-being and production [2,3]. Over the past decades, the average life span of humankind significantly improved, and poverty alleviated in some areas. At the same time, adequate knowledge of ES and understanding of scientific principles of effective management of ES are still lacking. For example, the excessive pursuit of provisioning services, approximately 60% (15 out of 24) of the ES are being degraded in the Millennium Ecosystem Assessment [4]. The degradation and loss of ES have a negative impact on human well-being, and directly threaten regional, national, and global ecological security [5].

The nonlinear relationship between different ES and complex patterns of their utilization by humans are common [6], which often makes ES dependent on each other and interact in multiple ways, displaying both trade-offs and synergies [7]. Trade-offs, or negative relationships, refer to an increase in the supply of one ES that leads to the reduction in the supply of another ES [8]. On the contrary, synergies, or positive relationships, mean that the supply of multiple ES increases or decreases simultaneously [5]. Due to the existence of such relationships, unexpected outcomes of ecosystem management practice are common. For example, the increase of provisioning services (grain, wood, etc.) may lead to the reduction of regulating services (nutrient cycling, soil conservation,

etc.) [9]. As another example, an increase in soil retention would improve soil carbon sequestration capacity [10]. Some European countries tried to encourage farmers to adopt more environmentally friendly cultivation techniques by offering financial incentives. However, these measures had no effect on the conservation of endangered species [11]. Therefore, consideration of trade-offs and synergies between ES is crucial for landscape planning and land management. It will help avoid costly adverse effects, and promote multi-functionality [12].

Influenced by natural and anthropogenic drivers, relationships between ES change spatially and temporally. Studies have analyzed the common drivers that underlie changes in ES, including climate change, land use changes, as well as policy interventions [13–15]. For example, the transformation of cultivated land into shrub and grassland enhanced soil conservation (SC) and carbon sequestration capacity but reduced water yield (WY) in the Loess Plateau, China [16]. In northern Shaanxi, SC and net primary productivity (NPP) had variable synergistic relationships in different land use types [10]. Finally, in the Yanhe river, SC, water retention (WR), and WY changed at sub-basin scale after the implementation of the Grain to Green Program (GTGP) [17]. Moreover, ES relationships occur at a different spatial and temporal scale [8], which increases uncertainties to be managed [18]. Previous studies have documented trade-offs and synergies among multiple ES [19–23]. However, they were mainly focused on single time periods and did not consider temporal changes [24]. Few studies have recently focused on changes in those relationships at different temporal scales. For example, simulations of ES relationships for the period of 2001−2070 revealed changes in some of those relationships involving regulating services [24]. A case study of Switzerland focused on a ten-year period assessment of ES and revealed robustness of the determined relationships between regulating and cultural services [25]. Although quantitative testing of ES and their relationships across temporal scales have been conducted recently, more evidence is needed to understand their dynamics and elucidate mechanisms underlying changes in ES relationships in a longer term to sustainably manage multiple ES.

In this study, we attempted to: (1) describe the spatial distribution and spatial trends of 5 ES indicators (i.e., livestock breeding (SHEEP); grain production (GRAIN); NPP; sandstorm prevention (SP); and WR) from 1979 to 2016 in the region of Uxin; (2) quantify changes in ES relationships over the same period; and to (3) reveal the driving factors behind change in the ES and their relationships.

## 2. Materials and Methods

### 2.1. The Study Area

Uxin is located in the southeast part of the Ordos Plateau in Inner Mongolia, North China. It has an average altitude of 1300 m, and extends in latitude from 37°38′ N to 39°23′ N and in longitude from 108°17′ E to 109°40′ E. Uxin has a typical temperate continental climate, with a mean annual precipitation of about 350 mm, a mean annual evaporation of 2200 mm, and a mean annual temperature of 6.8 °C. Fixed and moving sand dunes cover the majority of its landscape. Aeolian sandy soils and kastanozems are the most common soil types. Shrubs and subshrubs are the dominant vegetation type (e.g., *Caragana intermedia* and *Artemisia ordosica*). As a typical agro-pastoral transitional zone of northern China, the main land use is livestock husbandry and farming (Figure 1).

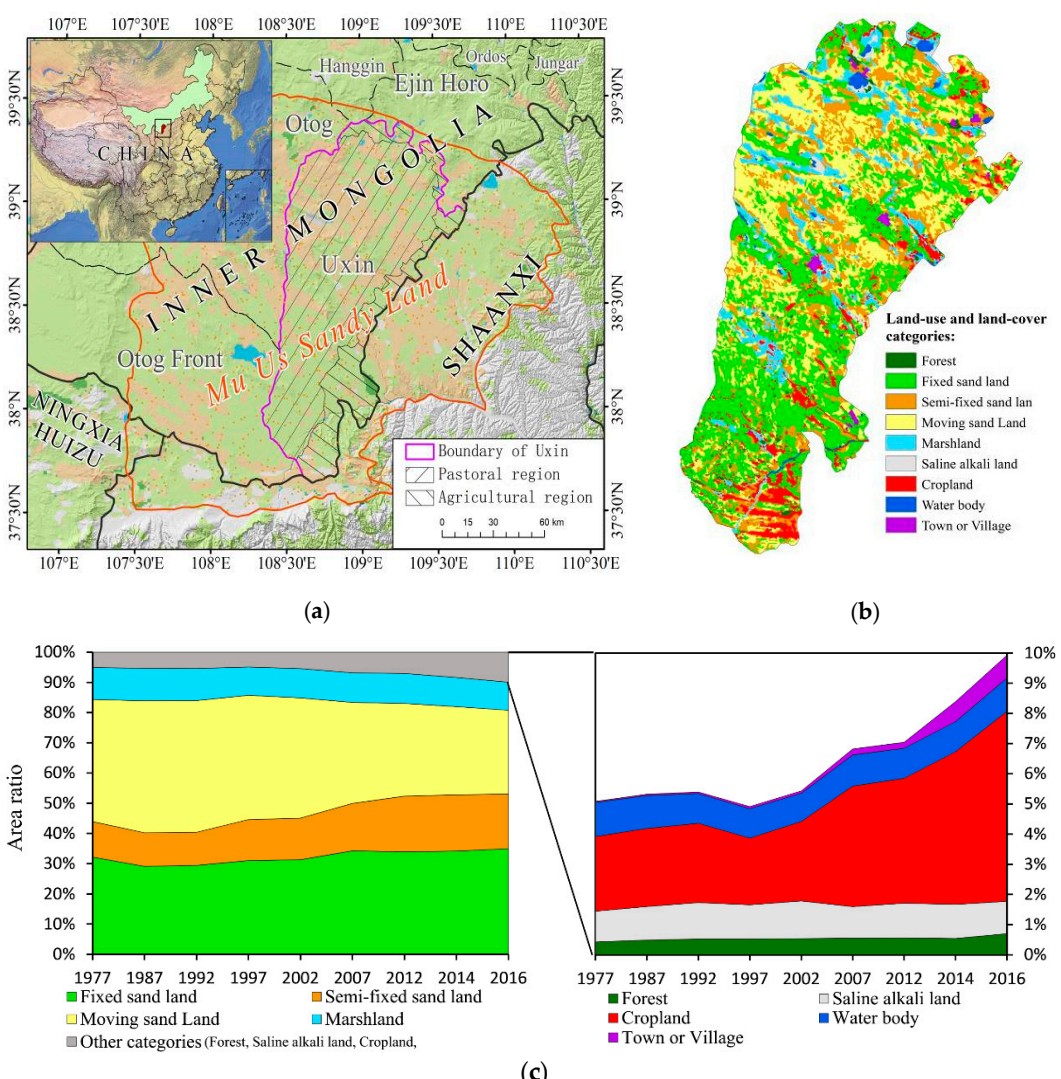

**Figure 1.** (**a**) Map of the case study area, Uxin is located in the southwestern part of Inner Mongolia, China. (**b**) LULC map in 2016 and (**c**) LULC change in Uxin in the period 1979–2016.

## 2.2. Quantification of Ecosystem Services

We investigated 5 ES indicators, including livestock breeding (SHEEP), grain production (GRAIN), net primary productivity (NPP), sandstorm prevention (SP), and water retention (WR). The selection criteria included the following. ES should: (1) be relevant to the well-being of local households and surrounding ecosystem; (2) has available remotely sensed data for spatially explicit monitoring of ES; and (3) require data that were available for long-term temporal ES relationship assessment. SHEEP and GRAIN are important to human well-being of local households. NPP reflects the growth status of vegetation. SP helps to reduce sandstorm disasters for Uxin and downwind areas (e.g., Xi'an city, Shaanxi Province, etc.). WR plays an important role in ecosystem vegetation recovery and household livelihoods. We mapped the annual value layer for each indicator from 1979 to 2016, which was then used to describe changes in ES. All data were resampled to a spatial resolution of 250 m (The results of 5 ES are available in Supplementary Materials).

### 2.2.1. Data Sources

Table 1 shows the data used to assess the 5 ES indicators. Climate data, such as mean temperature, precipitation, and wind speed, were interpolated using ArcGIS 10.3. Based on data availability, we used

land use and land cover (LULC) maps of the years 1978, 1987, 1992, 1997, 2002, 2007, 2012, 2014, and 2016 to represent the conditions for the following intervals: 1979–1987, 1987–1992, 1992–1997, 1997–2002, 2002–2007, 2007–2012, 2012–2014, and 2014–2016.

**Table 1.** Description and sources of the data used to assess the 5 ecosystem services (ES) indicators.

| Data | Data Description | Data Source |
|---|---|---|
| Climate data | Daily mean temperature; daily mean wind speed; daily rainfall | China Meteorological Data Service Center (http://data.cma.cn/) |
| Normalized Difference Vegetation Index (NDVI) | NOAA/AVHRR NDVI at 2000 m spatial resolution (1979–2001); MODIS MOD13Q1 NDVI at 250 m spatial resolution (2000–2016) | Chinese Academy of Agricultural Sciences; The Level-1 and Atmosphere Archive & Distribution System (LAADS) Distributed Active Archive Center (https://ladsweb.modaps.eosdis.nasa.gov/search) |
| Soil data | Sand fraction; silt fraction; clay fraction; organic carbon; calcium carbonate; bulk density | Harmonized World Soil Database V1.2 (http://www.fao.org/soils-portal/soil-survey/soil-maps-and-databases/harmonized-world-soil-database-v12/en/) |
| DEM | SRTM 90 m Digital Elevation Data Digital Elevation Model | The CGIAR Consortium for Spatial Information (srtm.csi.cgiar.org) |
| Land use and land cover (LULC)/vegetation map | Landsat 3 MSS at 90 m spatial resolution (1978); Landsat 5 TM at 30 m spatial resolution (1987, 1992, 1997, 2002, 2007); HJ-1B at 30 m spatial resolution (2012); Landsat 8 OLI at 30 m spatial resolution (2014, 2016) | LULC was classified into 9 categories; vegetation was classified into 6 categories and 16 sub-categories (the details are shown in Table 2). Those layers were visually interpreted and digitized on screen in ArcGIS 10.3 |

### 2.2.2. Livestock Breeding (SHEEP) and Grain Production (GRAIN)

SHEEP and GRAIN data were acquired from statistical yearbooks of Uxin for the period 1979–2016. The SHEEP is expressed in sheep units, where 1 cattle or horse equals to 4.5 sheep units [26]. In order to convert the aggregated, or non-spatial, statistical data into a spatial layer, the normalized difference vegetation index (NDVI) from remotely sensed satellite data was selected as a proxy to characterize the capacity of an ecosystem to provide forage. The density of sheep units is dependent on accumulated NDVI values as follows:

$$dSU_i = SU_i / NDVI_{i,sum}, \tag{1}$$

$$SU_{i,spati} = dSU_i * NDVI_{i,max}, \tag{2}$$

where $SU_i$, a vector map with map unit attributes linked to livestock numbers in 6 counties of Uxin. $NDVI_i$, sum was aggregated within the boundaries of each county for the year $i$. $NDVI_{i, max}$ is the maximum value component (MVC) for the year $i$. $SU_{i, spati}$ is the spatial layer of the livestock number for the year $i$.

The method of preparing spatial data for the GRAIN was similar to the one used for the livestock number. In this paper, farmland is extracted according to the NDVI threshold of 0.34. With the ArcGIS Zonal Statistics tool [27], this statistic mean value is calculated for each cropland zone defined by a zone dataset, based on values from NDVI dataset (a value raster). This method is consistent with the method of spatial representation of livestock number.

### 2.2.3. Net Primary Productivity (NPP)

The Carnegie-Ames-Stanford Approach (CASA) model was used to calculate the NPP [28]. This model takes into account the effects of solar radiation, temperature, and water stress on NPP.

$$NPP(x,t) = APAR(x,t) \times \varepsilon(x,t), \tag{3}$$

where *APAR(x, t)* is the vegetation photosynthetic effective radiation ($g\ C \cdot m^{-2} \cdot month^{-1}$) absorbed by grid *x* at month *t*; and $\varepsilon(x, t)$ is the actual utilization rate of light energy of vegetation (*g C*). *APAR(x, t)* was calculated using the following equation:

$$APAR(x,t) = SOL(x,t) \times FPAR(x,t) \times 0.5, \tag{4}$$

where *SOL(x, t)* is the total amount of solar radiation ($MJ \cdot month^{-1}$) at grid *x* for a specific month *t* [29]; 0.5 refers to the proportion of the solar effective radiation (the wavelength is 0.4–0.7 μm) to the total solar radiation; *FPAR(x, t)* is the absorption ratio of vegetation photosynthetic effective radiation. The utilization rate of light energy by vegetation is mainly affected by temperature and water conditions and calculated as follows:

$$\varepsilon(x,t) = T_{\varepsilon 1}(x,t) \times T_{\varepsilon 2}(x,t) \times W_{\varepsilon}(x,t) \times \varepsilon_{\max}, \tag{5}$$

where $\varepsilon_{max}$ represents the maximum light energy utilization ($g\ C \cdot MJ^{-1}$); $T_{\varepsilon 1}(x, t)$ and $T_{\varepsilon 2}(x, t)$ indicate the stress effects of the lowest and the highest temperatures on the utilization of light energy by vegetation; and $W_{\varepsilon}(x, t)$ is the influence coefficient of water stress, which was set to 0.542 $gC \cdot MJ^{-1}$ for the case study area [30].

### 2.2.4. Sandstorm Prevention (SP)

Based on the Revised Wind Erosion Equation (RWEQ) [31], sandstorm prevention (*SP*) can be estimated as the difference between potential wind erosion ($S_{Lp}$) and actual wind erosion ($S_L$). The formulas are as follows:

$$SP = S_{L,P} - S_L, \tag{6}$$

$$S_L = \frac{2z}{S^2} Q_{MAX} e^{-(z/S)^2}, \tag{7}$$

$$S = 150.71 (WF \times EF \times SCF \times K' \times C)^{-0.3711}, \tag{8}$$

$$Q_{MAX} = 109.8 (WF \times EF \times SCF \times K' \times C), \tag{9}$$

$$S_{Lp} = \frac{2z}{S_P^2} Q_{MAXp} e^{-(z/S_P)^2}, \tag{10}$$

$$S_P = 150.71 (WF \times EF \times SCF \times K')^{-0.3711}, \tag{11}$$

$$Q_{MAXp} = 109.8 (WF \times EF \times SCF \times K'), \tag{12}$$

where $S_L$ ($t \cdot km^{-2} \cdot a^{-1}$) is the actual soil loss caused by wind erosion; $S_{Lp}$ ($t \cdot km^{-2} \cdot a^{-1}$) is the potential soil loss; *QMAX* ($kg \cdot m^{-1}$) is the maximum transfer capacity; *S* (*m*) is the critical field length, defined as the distance at 63% of QMAX; *z* (*m*) is the maximum wind erosion distance, set to 50 m for the study area; *WF* ($kg \cdot m^{-1}$) is the climate factor, influenced by soil moisture, wind speed, and snow cover; *EF* is the soil erodibility factor; *SCF* represents soil crusting; *C* is the vegetation cover factor; and *K'* is the surface roughness factor. For the detailed formulas of RWEQ, see Jiang, et al. [32].

**Table 2.** The vegetation/land-use and land-cover categories.

| Vegetation Categories Level I | Vegetation Categories Level II | Land-use and Land-Cover Categories | Area Ratio of 2016(%) | Maximum Root Depth (mm) | *Kc* |
|---|---|---|---|---|---|
| Forest vegetation | Artificial forest | Forest | 0.70% | 3000 | 0.2993 |
| Shrubs and herbaceous vegetation in sandy land | *Artemisia ordosica* community on fixed sandy land | Fixed sand land | 27.80% | 600 | 0.2073 |
| | *Sabina vulgaris* community | | 2.46% | 2000 | 0.2373 |
| | *Caragana intermedia* Kuang et H.C.Fu and *A. ordosica* community | | 4.44% | 2000 | 0.2236 |
| | *A. ordosica, Sophora alopecuroides* L., *Cynanchum hancockianum* (Maxim.) Al. Iljinski. community | | 0.24% | 600 | 0.2004 |
| | *Salix cheilophila, S. psammophila* community | Semi-fixed sand land | 7.77% | 1000 | 0.2220 |
| | *Artemisia ordosica* community on semi-fixed sandy land | | 9.65% | 600 | 0.2034 |
| | *C. hancockianum* (Maxim.) Al., *S. alopecuroides* L., Iljinski., *Agriophyllum squarrosum* (Linn.) Moq. and *A. ordosica* community | | 0.67% | 300 | 0.2722 |
| | Pioneer community on moving sand Land | Moving sand Land | 27.74% | 200 | 0.1941 |
| Meadow and marshes | *Carex duriuscula* C.A.Mey. community *Achnatherum splendens* community | Marshland | 7.17% 2.13% | 200 300 | 0.2149 0.2459 |
| Halophyte vegetation | *Suaeda glauca* (Bunge) Bunge and *Salicornia europaea* community *Kalidium foliatum* (Pall.) Moq. and *Nitraria sibirica* Pall community | Saline alkali land | 1.04% 0.04% | 300 1000 | 0.3104 0.6047 |
| Agricultural vegetation | Cropland | Cropland | 6.29% | 300 | 0.3991 |
| Others | Water body | Water body | 1.11% | 0 | 0.6446 |
| | Town or Village | Town or Village | 0.75% | 0 | 0.2083 |

### 2.2.5. Water Retention (WR)

Considering factors such as topography, soil thickness, and permeability, WR was estimated based on the principle of water balance at watershed scale [33] as follows:

$$WR = Min\left(1, \frac{249}{Velocity}\right) \times Min(1, 0.3TI) \times Min\left(1, \frac{K_{sat}}{300}\right) \times WY, \tag{13}$$

where *WR* is the average water retention (*mm*); Velocity is the flow coefficient [29]; *TI* is the topographic index; $K_{sat}$ is the soil saturated hydraulic conductivity (cm·d$^{-1}$) [34]; and *WY* is the water yield of the basin (mm), calculated by InVEST 3.1.0 [33]. It is calculated as follows:

$$Yield_{xj} = \left(1 - AET_{xj}/P_x\right)P_x, \tag{14}$$

where $Yield_{xj}$ (*mm*) is the annual water yield for LULC type *j* in a given grid *x*; $P_x$ (*mm*) is the annual precipitation of grid cell *x*; and $AET_{xj}$ is the annual actual evapotranspiration (*mm*) of the LULC type *j* in the grid x. The $AET_{xj}$ was calculated using the following equations:

$$AET_{xj}/P_x = 1 + PET_{xj}/P_x - \left[1 + \left(PET_{xj}/P_x\right)^{\omega_x}\right]^{1/\omega_x}, \tag{15}$$

$$\omega_x = Z(AWC_x/P_x) + 1.25, \tag{16}$$

where $\omega_x$ is the parameter of natural climate and soil properties, defined as the ratio of annual water demand for plants and precipitation; $PET_{xj}$ is the annual potential evapotranspiration of LULC type *j* in grid *x* [35]; *Z* is the seasonal parameter for seasonal rainfall distribution, which was set to 11.54 for

the study area [36]; $AWC_x$ (mm) is the available water for plants in grid $x$ [37]. In addition, the InVEST water yield model also requires tabulated biophysical parameters including the maximum root depth of the vegetation of each LULC type and the evapotranspiration coefficient ($K_c$). The details are shown in Table 2.

*2.3. Statistical Analysis*

2.3.1. Trend Analysis of ES and Their Relationships

An image trend analysis was conducted to analyze ES changes in the period 1979–2016 in Uxin using the following equation:

$$b = \frac{l_{XY}}{l_{XX}}, \tag{17}$$

$$l_{XY} = \sum_{k}^{n} x_k y_k - \sum_{k}^{n} x_k \sum_{k}^{n} y_k / n, \tag{18}$$

$$l_{XX} = \sum_{k}^{n} x_k^2 - \left(\sum_{k}^{n} x_k\right)^2 / n, \tag{19}$$

where, $y_k$ represents a specific ES layer in year $k$ ($k$ = 1979, 1980 ... , 2016, $n$ = 38 years). $x_k$ is a grid layer; each grid cell is set to the same value for the year $k$; $b$ is the trend slope layer; A grid cell with $b < 0$ indicates a decreasing trend in ES in the period analyzed, while a grid cell with $b > 0$ indicates an increasing trend. The Arcpy scripts for trend analysis is available in Supplementary Materials.

ES relationships can be modeled by the Pearson correlation coefficient [18]. For some services, a positive correlation indicates a synergy, while a negative correlation represents a trade-off. It is calculated as follows:

$$r = \frac{l_{XY}}{\sqrt{l_{XX}l_{YY}}}, \tag{20}$$

$$l_{YY} = \sum_{k}^{n} y_k^2 - \left(\sum_{k}^{n} y_k\right)^2 / n, \tag{21}$$

where, $r$ is correlation coefficient layer of an ES pair. In spatial distribution characteristics of trade-offs analysis, $x_k$ and $y_k$ represent 5 ES layers in year $k$ ($k$ = 1979, 1980 ... , 2016, $n$ = 38 years). In temporal dynamics of ES trade-offs analysis, $x_k$ and $y_k$ represent k$^{th}$ pixel value of each ES pair in a specific year (e.g., 1979, 1980 ... , 2016).

The significance of $b$ value and $r$ were assessed by t-test using 2-tail t-distribution at 95% confidence interval.

2.3.2. Driving Forces of Ecosystem Services Interactions

Climate change, humans activities, and technological progress are main drivers of quantity and quality changes in ES [38]. We hypothesize 5 ES indicators are driven by climate factors and land use and indirectly influenced by technical progress and population growth, which might result in spatiotemporal changes in relationships of 5 ES. Land cover transformations driven by economy and population growth on one hand and technological progress, on the other, change the demand for natural resources and affect consumption patterns of different ES. To test this hypothesis, 7 independent variables were selected to perform a stepwise regression (Table 3), which minimizes the problem of multi-collinearity in variables. A max-min standardization was performed for all variables; the statistical analysis was performed using R 3.5.0 [39]. The full reproducible R code and data are available in Supplementary Materials.

**Table 3.** The driving forces of ecosystem services relationships.

| Variable Type | Variable Name | Abbreviation | Unit | Description |
|---|---|---|---|---|
| Climate change | Growing season precipitation | PRCP | mm | Cumulative value of precipitation during 3–10 months |
| | Cumulative temperature | TEM | °C | Cumulative value of annual temperature over 10 °C |
| | Cumulative wind speed | WIN | $m \cdot s^{-1}$ | Annual cumulative value of wind speed over 5 m/s |
| Land use changes | Land use intensity change | FAM_LUI | – | Equal to grain yield divided by sown area of each year. The greater the value, the greater the intensity of land use, representing the improvement of the intensification level of crop planting |
| | Grazing pressure | GRS_PRS | – | Equal to the annual number of breeding livestock divided by the annual NDVI cumulative value (minus the area of cultivated land). The greater the value, the greater the grazing pressure of natural grassland utilization, which may result in grassland degradation |
| Technical progress | Total mechanical power of agriculture and animal husbandry | AGMACH | $\times 10^4$ KW | Mechanical input in crop farming and irrigation for artificial grassland, and in animal husbandry |
| Population change | Total population | POP | People | The demand for ecosystem services from population growth |

## 3. Results

### 3.1. Trends in Ecosystem Services

#### 3.1.1. SHEEP

SHEEP increased in 38 years, from $7.64 \times 10^5$ sheep units (*SU*) in 1979 to $12.24 \times 10^5$ SU in 2016, with an annual increase of $1.21 \times 10^4$ SU (see Figure 2-A1). SHEEP showed a slow declining trend ($1.92 \times 10^5$ *SU*) from 1979 to 2001, and an increasing trend from 2002 to 2016, incrementing from $6.89 \times 10^5$ *SU* to a peak of $15.50 \times 10^5$ *SU* in 2007. Mean spatial SHEEP values for the 38-year period range from 8 to 12 *SU·hm*$^{-2}$, i.e., about 37% highest than the lowest values (Figure 2-A2). The region with the lowest values is in the northwest of Uxin ($< 3$ *SU·hm*$^{-2}$) while the northeast, central, and southern parts of Uxin are characterized by higher SHEEP values. About 71% of the total area, mainly in the northeast and west of the region exhibited an increasing trend in SHEEP values, with slopes (*b* value, Equation (12)) ranging from 0 to 0.75 (see Figure 2-A3). Decreasing trends in SHEEP values with slopes between –0.15 and 0 are found in 4.87% of the total area, mainly exhibiting scattered patterns.

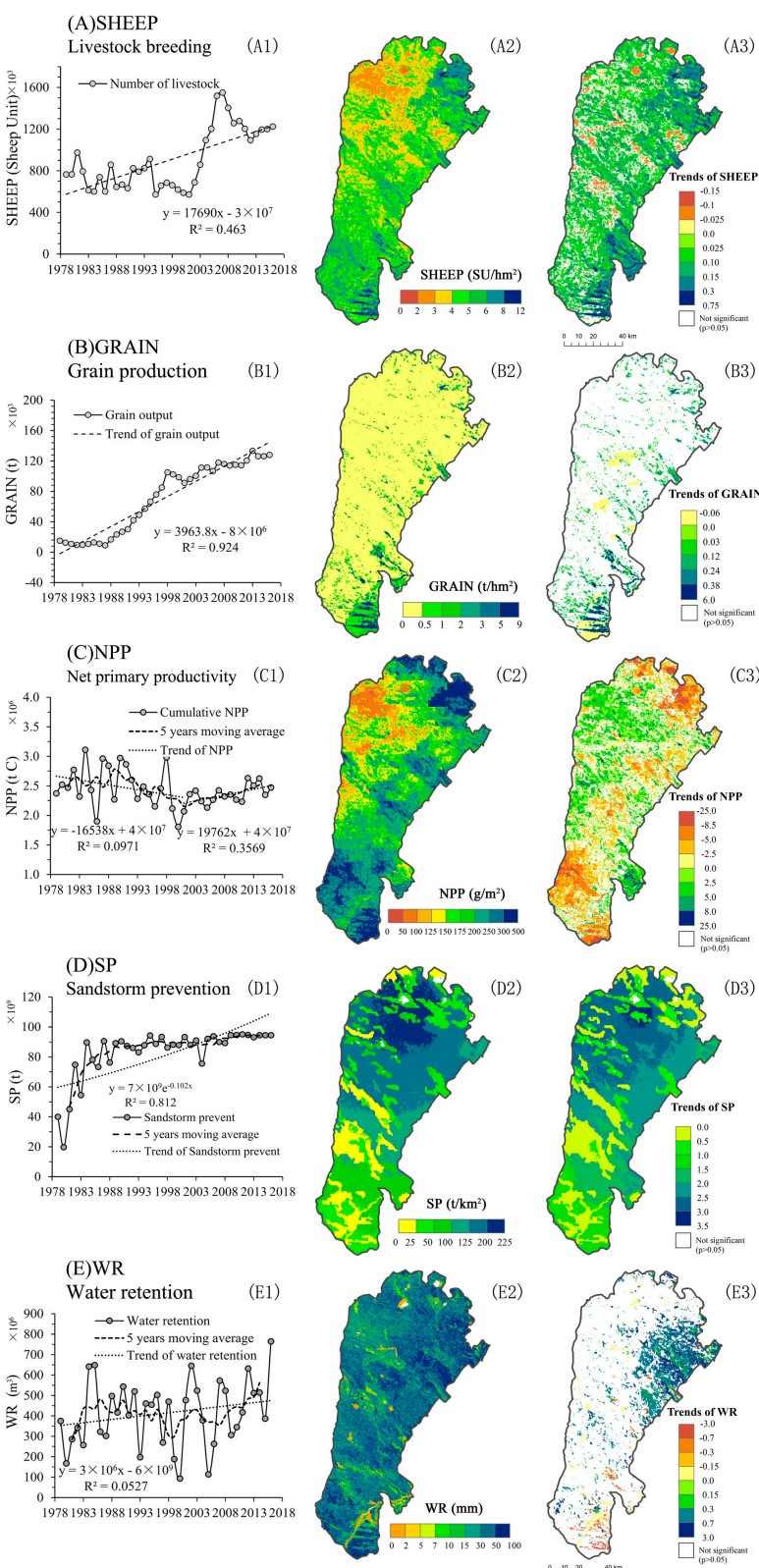

**Figure 2.** Maps of mean values and change trends of 5 ecosystem services (ES) from 1979 to 2016 in Uxin. A1-E1 represents the aggregated ES variation from 1979 to 2016; A2-E2 represents the mean 38-years value of ES; and A3-E3 represents the spatial distribution of the change trend in ES in the 38 years analyzed.

### 3.1.2. GRAIN

During the 38-year period GRAIN also increased 7.54 times from $1.50 \times 10^4$ $t$ to $1.28 \times 10^5$ $t$, with an average annual change of $2.98 \times 10^3$ $t$ (see Figure 2-B1). The highest value of GRAIN is found in about 10% of the total area, mainly in the southeastern, eastern, and northern parts of Uxin. Areas with lowest GRAIN value are scattered across the Uxin region and occupy about 4% of the total area (see Figure 2-B2). GRAIN exhibited an increasing trend (see Figure 2-B3), especially in the southern region, where its slope ranged from 0.38 to 0.60 (with an area ratio of 1.37%). The slope of the eastern and northern regions was between 0.03 and 0.38 (with an area ratio of 9.07%).

### 3.1.3. Net Primary Productivity

Two trends of NPP can be identified over the past 38 years (see Figure 2-C1). NPP decreased gradually during 1979–2000, from the maximum of $2.96 \times 10^6$ $t$ C in 1987 to $1.81 \times 10^6$ $t$ C in 2000, resulting in the total decrease of $1.16 \times 10^6$ $t$ C. In the period 2001–2016, it showed an increasing trend, with an average annual NPP of $2.35 \times 10^6$ $t$ C, indicating vegetation recovery in this area. Highest mean NPP ($300–500$ g·m$^{-2}$) is mainly observed in the north and south of Uxin, with an area ratio of 6.88%. These regions showed a declining trend in NPP (Figure 2-C2 and Figure 2-C3). Lowest mean NPP is found in the western and northwestern Uxin (9.03%). These areas had an increasing trend of NPP.

### 3.1.4. Sandstorm Prevention (SP)

SP exhibited an increasing trend (Figure 2-D1) during the 38-year period (the cumulative wind speed in Uxin is shown in Supplementary Materials), with the maximum value in 2010 ($9.47 \times 10^{10}$ t) and the lowest in 1980 ($1.97 \times 10^{10}$ t). The annual average SP was $8.35 \times 10^{10}$ t, which reduced the threat of sandstorm disasters in Uxin and downwind region of Mid-western China. Highest SP values are mainly located in the moving sand dune area in the northern part of Uxin (with a SP higher than 200 t·km$^{-2}$). Lowest SP values are mainly found in the south of Uxin (Figure 2-D2). SP showed a consistent increasing trend its distribution characteristics were similar to its spatial distribution (see Figure 2-D3).

### 3.1.5. Water Retention

Different from the study of similar region in Xilin River Basin [36], WR showed an increasing trend in the period 1979–2016, with its rate changing from $3.75 \times 10^8$ m$^3$ in 1979 to $7.65 \times 10^8$ m$^3$ in 2016 resulting in an annual change of $1.03 \times 10^7$ m$^3$ (see Figure 2-E1). The mean WR was greater than 15 mm in more than 90% of the total region area, mainly in the east and south of Uxin (see Figure 2-E2). About 24.16% of the total region area had mean WR in the range of 15–30mm; 49.54% had mean WR of 30–50 mm; and 16.82% had mean WR of 50–100 mm. Over 16.33% of Uxin, mainly the northeastern area, showed a slight increasing trend for WR. Only 1.88% of the total region, areas scattered the south and central part, showed a decreasing trend for WR (see Figure 2-E3). Water is the main limiting factor of vegetation growth and agricultural production in this area. With WR increased gradually, there will be enough water resources to meet the demand of agricultural expansion and population growth, and indirectly reducing vegetation degradation.

### 3.2. Change Trajectories of ES Relationships

### 3.2.1. Temporal Dynamics of ES Relationships

Trends in ES relationships varied in the 38 years analyzed (Figure 3, the bottom-right triangle diagram matrix). Positive correlations between SHEEP–GRAIN, SHEEP–NPP, and GRAIN–NPP have been gradually strengthening, while negative correlations among some ES pairs, such as SHEEP–SP, GRAIN–SP, and NPP–SP, displayed a weakening trend during the study period. Yet some positive

correlations, such as for SHEEP–WR, GRAIN–WR and NPP–WR, switched into negative correlations. The SP–WR pair is characterized by almost an equal number of positive and negative correlation.

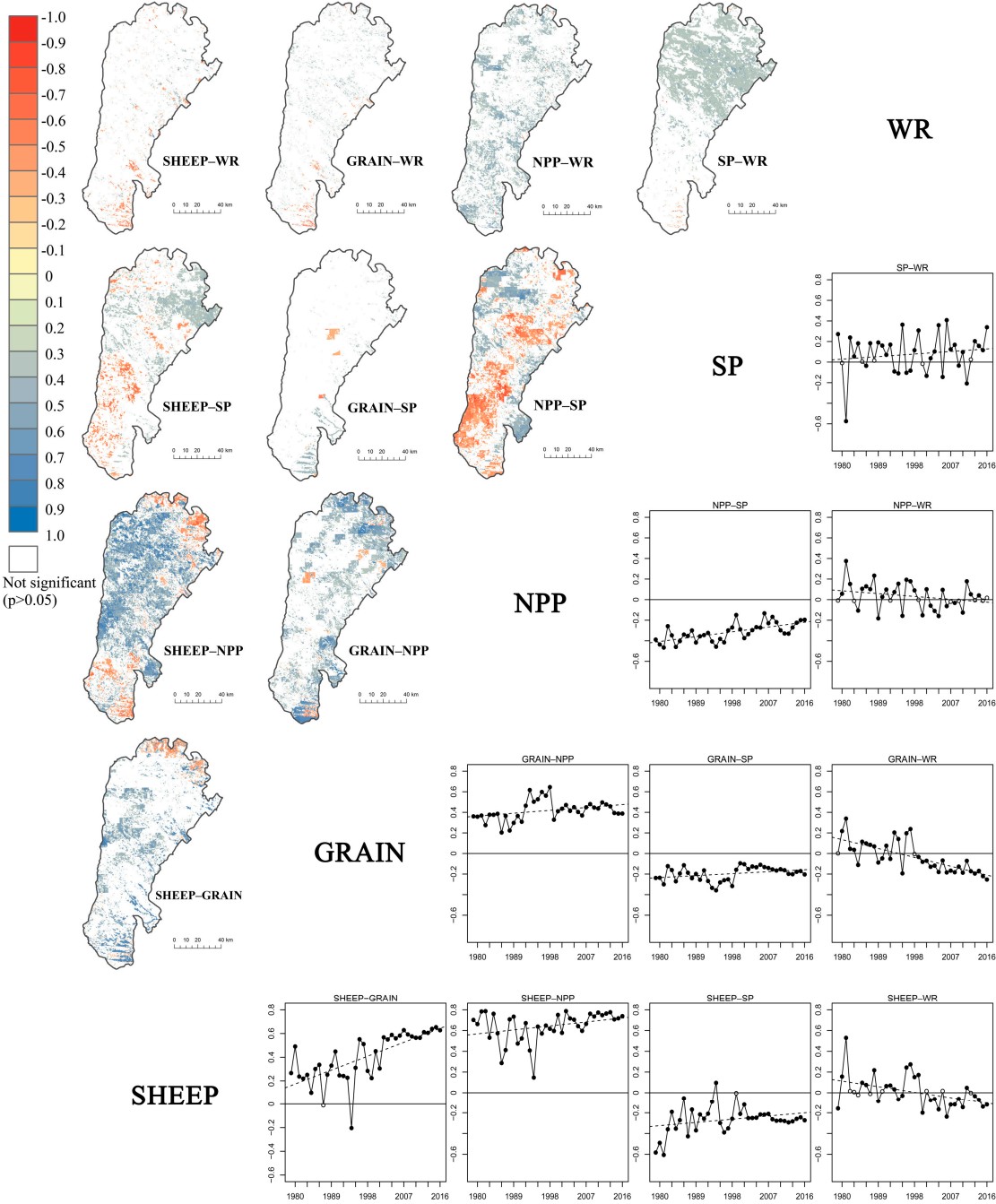

**Figure 3.** Maps of spatial distribution and change trajectories of relationship between ecosystem services from 1979 to 2016 in Uxin. (1) SHEEP (livestock breeding); GRAIN (grain production); SC (soil conservation); WR (water retention); and NPP (net primary productivity). (2) The bottom-right triangle diagram matrices represent the dynamic of ES relationship in the 38 years analyzed. The solid circle, or hollow circle, indicates that the relationships between two services passed (or not) the 2-tailed t-test at significance level $\alpha < 0.05$. (3) The top-left triangle diagram matrices show the spatial distribution of ES relationships in 38 years. The color red and blue indicate negative to positive relationship. The deeper the color, the stronger the relationship.



3.2.2. Spatial Patterns of ES Relationships

According to the spatial patterns of relationships between various ES (Figure 3, the top-left triangle diagram matrices), the pairs GRAIN–NPP, NPP–WR and SP–WR exhibited almost uniform positive correlation across the whole region, while SHEEP–GRAIN, SHEEP–NPP, SHEEP–SP and NPP–SP exhibited a spatial clustering in positive and negative correlations. Finally, no specific patterns are observable for other pairs of ES.

*3.3. Driving Forces of ES Relationship Changes*

We used the stepwise regression to analyze the driving forces of change in the 10 relationships among ES pairs (see Table 4). In the period 1979–2016, SHEEP–GRAIN showed a positive relationship with GRS_PRS and AGMACH, suggesting that both the increased grazing pressure and the investments in the mechanical power of agriculture and animal husbandry promoted an improvement of the synergies with SHEEP–GRAIN. SHEEP–NPP also showed a positive relationship with GRS_PRS and AGMACH, while the increase of POP tended to inhibit the increase of these two synergy relationships. SHEEP–SP was negatively correlated with WIN and AGMACH. SHEEP–WR had a negative correlation with AGMACH, indicating that the increase in investments in artificial grassland by herdsmen were accompanied by an increase in sheep number and by an excessive consumption of groundwater, which tended to decrease their synergy relationships. GRAIN–NPP had a positive correlation with TEM, whereby increases in cumulative temperature tended to increase the synergies with GRAIN–NPP. GRAIN–SP was positively correlated with TEM, while GRAIN–WR was negatively correlated with changes in AGMACH. NPP–SP had a significantly positive correlation with TEM. NPP–WR was negatively correlated with GRS_PRS. SP–WR was positively correlated with WIN, GRS_PRS, and POP, and was negatively correlated with AGMACH.

*3.4. Temporal Dynamics of ES Relationships*

The temporal dynamics of the trade-offs and synergies among ES reflected the responses to social and environmental changes [24]. Furthermore, this study also found that both trade-offs and synergies were not invariable in the long period. On the contrary, they were likely influenced by a variety of drivers, such as climate, human activities, and technological progress. According to the results of the stepwise regression (see Table 4), there are four possible types of relationship between driving forces and ES pairs (see Figure 4). The direct and indirect driving forces of the relationship between ecosystem service pairs are shown in Table 5. The four types of relationships between driving forces and ES pairs are briefly described below.

(i) Driving factors directly affect two ES, increasing or decreasing the provision of ES, and resulting in changes in trade-offs and synergies (see Figure 4a). In this study we showed the presence of a synergistic relationship between SHEEP and GRAIN (see Figure 2, bottom-right part), which followed an increasing trend over the 38 years analyzed. The stepwise regression analysis showed that SHEEP and GRAIN were positively affected by technical progress factors. The mechanical input in crop farming and cultivated land during the last three decades led to an increase in food production by farm families. Triggered by the prohibition of open grazing policy since the early 2000s [40], herdsmen families increased their investments in irrigation for artificial pasture and fenced grazing, resulting in an increase in sheep breeding. Water is the main limiting factor of vegetation growth and agricultural production in this area. This study showed that the synergies between GRAIN–WR were transformed into trade-offs in the 38 years analyzed (see Figure 3), and were negatively affected by technical progress factors (see Table 4). By the late 1990s, when the mechanical input in crop farming increased, the water consumption by intensive agriculture increased gradually, while the WR in the region did not increase significantly (see Figure 2). Therefore, the synergy between GRAIN–WR gradually changed into a trade-off. Like GRAIN–WR, SHEEP–WR was negatively correlated with AGMACH, suggesting

that the mechanical input in artificial grassland directly increased SHEEP provision and an excessive consumption of groundwater, tending to decrease their synergy relationship.

**Table 4.** Driving forces analysis of changes in ES relationships.

| Dependent Variable | Model | Coefficient | t Value | Pr(>\|t\|) | Model Summary |
|---|---|---|---|---|---|
| SHEEP–GRAIN | (Intercept) | 0.334 *** | 4.624 | 0.000 | $R^2 = 0.61$ |
| | GRS_PRS | 0.353 ** | 2.739 | 0.010 | F = 12.86 |
| | AGMACH | 0.605 * | 2.446 | 0.020 | $p = 0.00$ |
| | POP | −0.431 | −1.348 | 0.187 | |
| SHEEP–NPP | (Intercept) | 0.773 *** | 12.367 | 0.000 | $R^2 = 0.42$ |
| | GRS_PRS | 0.276 * | 2.470 | 0.019 | F = 5.95 |
| | AGMACH | 0.820 *** | 3.831 | 0.001 | $p = 0.00$ |
| | POP | −0.934 ** | −3.371 | 0.002 | |
| SHEEP–SP | (Intercept) | −0.022 | −0.427 | 0.672 | $R^2 = 0.46$ |
| | WIN | −0.534 *** | −5.366 | 0.000 | F = 14.70 |
| | AGMACH | −0.228 ** | −3.243 | 0.003 | $p = 0.00$ |
| SHEEP–WR | (Intercept) | 0.070 * | 2.270 | 0.029 | $R^2 = 0.19$ |
| | AGMACH | −0.195 ** | −2.921 | 0.006 | F = 8.53 |
| | | | | | $p = 0.01$ |
| GRAIN−NPP | (Intercept) | 0.339 *** | 14.042 | 0.000 | $R^2 = 0.326$ |
| | FAM_LUI | 0.211 *** | 3.800 | 0.001 | F = 8.46 |
| | AGMACH | −0.093 | −1.626 | 0.113 | $p = 0.00$ |
| GRAIN–SP | (Intercept) | −0.236 *** | −13.412 | 0.000 | $R^2 = 0.17$ |
| | TEM | 0.093 ** | 2.711 | 0.010 | F = 7.35 |
| | | | | | $p = 0.01$ |
| GRAIN–WR | (Intercept) | 0.149 *** | 3.217 | 0.003 | $R^2 = 0.57$ |
| | PRCP | −0.110 | −1.612 | 0.116 | F = 13.64 |
| | GRS_PRS | −0.122 | −1.677 | 0.103 | $p = 0.00$ |
| | AGMACH | −0.185 * | −2.180 | 0.036 | |
| NPP–SP | (Intercept) | −0.413 *** | −19.878 | 0.000 | |
| | TEM | 0.115 * | 2.222 | 0.033 | $R^2 = 0.61$ |
| | FAM_LUI | −0.127 | −1.505 | 0.142 | F = 12.84 |
| | GRS_PRS | 0.143 | 1.487 | 0.146 | $p = 0.00$ |
| | POP | 0.081 | 1.370 | 0.180 | |
| NPP–WR | (Intercept) | 0.088 *** | 2.729 | 0.010 | $R^2 = 0.11$ |
| | GRS_PRS | −0.105 * | −2.126 | 0.040 | F = 4.52 |
| | | | | | $p = 0.04$ |
| SP–WR | (Intercept) | −0.931 ** | −3.415 | 0.002 | |
| | WIN | 1.061 ** | 3.464 | 0.002 | $R^2 = 0.34$ |
| | FAM_LUI | −0.478 | −1.903 | 0.066 | F = 3.33 |
| | GRS_PRS | 0.610 * | 2.520 | 0.017 | $p = 0.02$ |
| | AGMACH | −1.379 ** | −3.151 | 0.004 | |
| | POP | 2.146 ** | 3.409 | 0.002 | |

Significance codes: *** $p < 0.001$; ** $p < 0.01$; * $p < 0.05$. The dependent variable $ES_1$–$ES_2$ represents the relationship (correlation coefficient) between two ES in the 38 years.

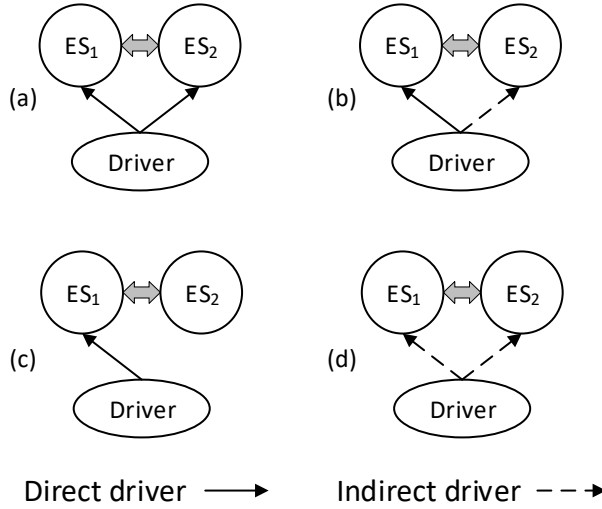

**Figure 4.** Relationships between driving forces and ES pairs. (**a**) driving factor directly affect two ES, (**b**) driving factor directly affect one ES and indirectly affect another ES, (**c**) driving factor directly affect one ES, (**d**) driving factor indirectly affect two ES. Ellipse represents driving factors, circle represents ES, and twin arrow represents relationships between ES.

**Table 5.** Direct and indirect driving forces of relationships among ES.

| Dependent Variable | Driving Force | Direct Driver | Indirect Driver |
|---|---|---|---|
| SHEEP–GRAIN | GRS_PRS | SHEEP | |
| | AGMACH | SHEEP, GRAIN | |
| SHEEP–NPP | GRS_PRS | SHEEP, NPP | |
| | AGMACH | SHEEP | NPP |
| | POP | SHEEP | NPP |
| SHEEP–SP | WIN | SP | |
| | AGMACH | SHEEP | SP |
| SHEEP–WR | AGMACH | SHEEP, WR | |
| GRAIN–NPP | FAM_LUI | GRAIN | |
| GRAIN–SP | TEM | GRAIN | SP |
| GRAIN–WR | AGMACH | GRAIN, WR | |
| NPP–SP | TEM | NPP | SP |
| NPP–WR | GRS_PRS | NPP | |
| SP–WR | WIN | SP | |
| | GRS_PRS | | SP, WR |
| | AGMACH | WR | SP |
| | POP | WR | SP |

(ii) Driving factors directly affect one ES and indirectly affect another ES, resulting in changes in trade-offs and synergies (see Figure 4b). SHEEP–NPP showed a negative relationship with POP. Over the 38 years investigated, population growth led to a direct increase in sheep supply and to an increase in water consumption, thereby indirectly inhibiting the increase of NPP. SHEEP–SP had a negative correlation with AGMACH (see Table 4), indicating that increased investments in artificial grassland by herdsmen were accompanied by an increase in sheep number and by an excessive consumption of groundwater, which prevented vegetation recovery, thereby indirectly decreasing SP. GRAIN–SP was positively correlated with TEM. Due to the extension of growing seasons thanks to climate warming, which increased both grain yield [41] and biomass (NPP) [42], vegetation restoration was promoted, which indirectly increased SP. Similarly, thanks to climate warming (TEM), NPP gradually increased,

which in turn led to changes in the supply of SP, resulting in the increase of the synergy relationship between NPP and SP during the period investigated. The increase in AGMACH resulted in an increase in the trade-offs between SP and WR. The increase of mechanical input in crop farming directly decreased the supply of WR, and had a negative impact on vegetation restoration in the surrounding ecosystems, which in turn indirectly reduced SP provision. SP–WR was positively correlated with POP. Population growth directly led to an increase in water use (i.e., a decrease in WR), and reduced the trend of the adverse impact of vegetation restoration on SP. SP and WR both decreased at the same time, thus increasing the synergy between SP and WR.

(iii) Driving factors directly affect one ES, while having at the same time a smaller effect on another ES. Changes in the supply capacity of only one ES, will eventually lead to changes in trade-offs and synergies (see Figure 4c). SHEEP–SP showed a positive relationship with WIN. The decrease in wind speed led to an increase in SP, which led to an increase in the synergistic relationship between them. FAM_LUI had a positive impact on GRAIN–NPP. Agricultural intensification promoted an increase in grain yield, thus increasing the synergistic relationship between them. The increase in grassland utilization (GRS_PRS) resulted in a decrease in NPP. Both WR, and the trade-offs between NPP and WR, recorded an increase in the 39 years investigated. SP–WR was positively correlated with WIN, indicating that the decrease of the average wind speed led to an increase in SP. Moreover, the increase in WR fostered an increase in the synergistic relationship between them.

(iv) Driving factors indirectly affect two ES, changing their supply rate and influencing their trade-offs and synergies (see Figure 4d). SP–WR was positively correlated with GRS_PRS, suggesting that the increase of grazing pressure on grassland directly reduced vegetation cover, thereby indirectly reducing SP and WR, and increasing the synergistic relationship between them.

## 4. Discussion

### 4.1. Trade-offs and Synergies, and ES Spatial Heterogeneity

Our results demonstrate that trade-offs and synergies among ES are spatially heterogeneous. Li et al. (2017) revealed that in grasslands, the relationships between ES tend to be in the form of synergies, while in built-up land and farmland they tend to be in the form of trade-offs [43]. In our research, SHEEP–GRAIN, SHEEP–NPP, SHEEP–SP, and NPP–SP exhibited a spatial clustering in positive and negative correlations. Land use conflicts were one of the causes of trade-offs and synergies between ES [12,44,45]. The heterogeneity of ES is usually caused by land use [46], whereby different types of land use provide different levels of ES [10]. For example, the spatially heterogeneous relationship between SHEEP and NPP, caused by their regional distribution (see Figure 2-A2 and C2), resulted in an imbalance in ES trade-offs. The area with the lowest SHEEP value is located in the northwestern part of Uxin, while the areas with a lower vegetation coverage showed a decrease trend (see Figure 2-A3). The areas with high SHEEP values are located in the northeastern and southern regions, and showed an increasing trend over the 38 years analyzed (see Figure 2-A3). NPP exhibited similar spatial patterns (see Figure 2-C2). However, it showed trends of change that are opposite to those of SHEEP (see Figure 2-C3). Xu et al. (2017) also found that, due to the presence of hot-spot areas of ES in Ningxia, which contains 37% of the cultivated land and produces 57% of grain, the provision and regulation services in the region did not appear to develop trade-off relationships [44].

### 4.2. Management Implications

Our research suggests that it is necessary to perform a long-term analysis of ES and their drivers, to better understand the trade-offs and synergies among them. Due to the different sensibility of ES to climate change and human disturbance, changes in ES relationships can be abrupt with high inter-annual precipitation variations [24]; hence, the ES were chosen at different time periods, and this may have influenced the results of the various ES interactions (see Figure 5). Hence, combined with the statistical test, the timely assessment and monitoring of ES and their relationships are needed to

robustly identify the change trends of each ES, to detect threshold or lag effects in ES interactions [43], to help avoid surprising trade-offs, and to take advantage of emerging synergies [24].

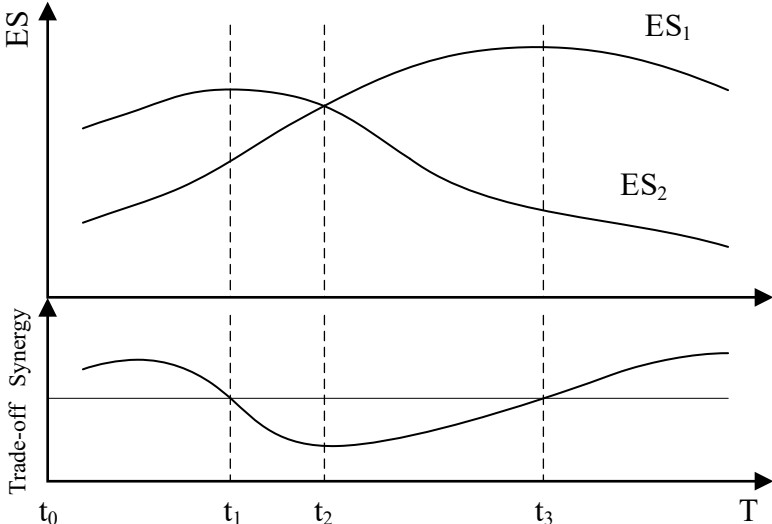

**Figure 5.** Changes in ES and in their interactions in terms of trade-offs or synergies (modified from [43]). $ES_1$ and $ES_2$ have a synergy relationship in the period $t_0$-$t_1$, a trade-off relationship in the period $t_1$-$t_2$ and $t_2$-$t_3$, and a synergy relationship after $t_3$.

According our study, the magnitude and direction of ES trade-offs and synergies may also vary, responding to changes in both natural and anthropogenic drivers. Our results suggest that more adaptive approaches for ecosystem management are required [47]. For example, revegetation in arid areas creates potentially conflicting demands for water between the ecosystem and humans [48]; these processes, although enhancing biomass storage or carbon sequestration, may decrease water availability [10,49]. In this case, the relationships between GRAIN–WR and NPP–WR will change from synergies into trade-offs. Under the warming and drying trend of regional climate [16], the continuous water demand by agriculture and humans may cause a decline in WR, and eventually result in vegetation degradation. Therefore, the implementation of a high-efficiency water saving agriculture (e.g., through drip irrigation or plastic film) is necessary to reduce water consumption rates and to improve cropland productivity, thereby improving other services as complementary or insurance measures for ecological rehabilitation [10].

According to our study, the inner relationship between driving forces should be cautious about. And relationship among drivers may display both trade-offs and synergies or interaction effect. For example, population variation and technical progress would affect the change of land use, and eventually lead to the change of ES supply in this region. Restricted by prohibition of open grazing policy since the early 2000s, herdsmen families in Uxin increased mechanical power in artificial pasture. Such changes in land management increased sheep breeding and reduced grazing pressure on nature grassland [40]. Farm families increased their investment in crop farming and cultivated land during the last three decades, which resulted in cropland expansion hot-spot areas and increase of grain production. It was founded in "Grain for Green" project region in Ningxia, China, the hot-spot areas of 37% of well managed land had contributed 57% of grain output [44]. Similarly, Pretty et al. (2006) found that farming systems adopting sustainability enhancing practices had increased productivity and improving the supply of ES [50].

### 4.3. Uncertainties in ES Assessment

Our study has some limitations and uncertainties. Input data accuracy is one of the sources of errors, such as those stemming from the 250-m MODIS data products and 1-km NOAA NDVI

data used in the NPP calculation process. We used a regression method to bridge the resolution gap between these two data sources. However, due to the high contrast of sand land when in proximity to other land covers, uncertainties are unavoidable when using NOAA data, which have a coarse spatial resolution. Our analysis showed a higher variability of NPP in the period 1979–1999 than in the period 2000–2016. The models used in this study also have some limitations. SP was based on the RWEQ model, which was developed in the 1980s for the Great Plains region in the USA. The use of recommended parameters in our study is another source of uncertainty. Therefore, measurement data should be acquired in future research, to provide a basis for the verification of parameters and results from the local application of this model. This should allow for a more accurate evaluation of ecosystem services in the sandy area of grassland.

## 5. Conclusions

The analysis of long-term trends in ES is important to understand their responses to climate change and human activities. In this paper, we analyzed the trends of ES, and the changes in trade-offs and synergies between ES and their driving forces, in the period 1979–2016. Our study highlights the importance of considering long-term trends of ES and their drivers in understanding ES interactions. Our results demonstrate that the magnitude of ES trade-offs and synergies vary in the long period, and may even switch for some pairs of ES. Trade-offs and synergies also showed regional distribution characteristics. The analysis of the driving forces of ES relationship change found that climate factors, land use change, technological progress, and population are the main factors behind changes in ES and in the relationships between them.

**Supplementary Materials:** The following are available online at http://www.mdpi.com/2071-1050/11/21/6041/s1, Data S1: Spatial distribution map of 5 ecosystem services from 1979 to 2016, R and Arcpy scripts, Climate change in Uxin, Driving factors of ecosystem services.

**Author Contributions:** Conceptualization, J.Z. and X.L.; Methodology, J.Z. and T.B.; Program scripts, J.Z. and X.Z.; Formal Analysis, J.Z. and T.B.; Writing-Original Draft Preparation, J.Z.; Writing—review & editing, A.B.

**Funding:** This research was funded by National Natural Sciences Foundation of China grant number 31500384 and 31500366, National Key Research and Development Program of China grant number 2016YFC0500707 and 2016YFC0500503, Fundamental Research Funds for the Central Universities (Program for ecology research group).

**Conflicts of Interest:** The authors declare no conflict of interest.

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
