# Peer review of "How Do Trade-Offs and Synergies between Ecosystem Services Change in the Long Period? The Case Study of Uxin, Inner Mongolia, China"

_sustainability, doi:10.3390/su11216041_

Round 1
Reviewer 1 Report
My comments have took into account in the new version.
I have not other comments.
Author Response
Reviewer have not other comments.
Reviewer 2 Report
General comments:
Good job on improving the Discussion and Conclusions sections of your manuscript.
I left a list of specific comments last time of which many seem to have been purely disregarded, which is a bit disappointing. Many of them are justified as they either refer to grammar issues, unclear statements or confusing sentences. I highly recommend addressing these comments to further improve the manuscript. I include below the list of comments that remain to be addressed satisfactorily.
Specific comments:
Introduction:
L36: 60%... worldwide? Still unclear. Please clarify.
L.53: The word “unpredictable” seems inappropriate here. If these phenomena were totally unpredictable, then there would be no real need to study them. Please rephrase.
Materials and methods:
L.86: “subshrubs” (plural)
L.92-93: These aren’t “standard” ES categories such as the ones defined in the MEA, TEEB or CICES classification systems. Therefore, you should further explain where these ES indicators fit compared to the standard ES classification schemes.
L.122: Briefly explain what the ArcGIS Zonal Statistics tool does and add a reference.
L.138: “0.542”. Why this value? What is the assumption here? Explain and add supporting reference.
L.168: Move title of Section 2.3 to next page
L.170: performed ==> conducted
L.187: quality of ES ==> quality changes in ES
Table 3: Why these 10°C and 5m/s thresholds? Explain your assumptions!
Results:
L.208: Spatially mean ==> mean spatial
L.211: highest ==> higher
L.221 and further: In which units are you expressing all your slope values? %? Degrees?
Figure 2: This figure is still hardly readable! Please expand to entire page.
L.232-234: NPP are ==> NPP is
L.236-241: These are huge numbers – are you sure of them? What do these observations tell you about the local context and ES provision in general? What would be the detrimental implications of not having all this amount of sand trapped by ecosystems?
L.243-250: A bit strange to express water volumes in tonnes instead of m3. Any explanation? Also, interpret a little bit more. How do these observations differ from the rest of China? What are the implications? Are these numbers good or bad? What is the impact on ecosystems?
Figure 3: Expand figure to improve readability.
Table 4: Add a column that briefly describes each variable to help the reader.
Discussion:
L.333: positive ==> positively
Section 4.1: I reckon that this section is very descriptive and belongs more to the Results part than the Discussion. It seems to me that you really start to “discuss” your results in Section 4.2.
L.385: tradeoffs ==> trade-offs
Author Response
Response to Reviewer 2 Comments
Point 1: L36: 60%... worldwide? Still unclear. Please clarify.
Response 1:
We rewrote this unclear sentence.
Please see L35-36. “For example, the excessive pursuit of provisioning services, approximately 60% (15 out of 24) of the ecosystem services are being degraded in Millennium Ecosystem Assessment.”
Point 2: L.53: The word “unpredictable” seems inappropriate here. If these phenomena were totally unpredictable, then there would be no real need to study them. Please rephrase.
Response 2:
Yes, the word “unpredictable” is not inappropriate. We deleted this sentence and reorganized this paragraph.
Point 3: L.92-93: These aren’t “standard” ES categories such as the ones defined in the MEA, TEEB or CICES classification systems. Therefore, you should further explain where these ES indicators fit compared to the standard ES classification schemes.
Response 3:
We selected 5 ES indicators: livestock breeding (SHEEP), grain production (GRAIN), net primary productivity (NPP), sandstorm prevention (SP), and water retention (WR).
Although, they aren’t “standard” ES categories, they are important to local households and surrounding ecosystem in this region. SHEEP and GRAIN are important to human well-being of local households. NPP reflects the growth status of vegetation. SP helps to reduce sandstorm for Uxin and downwind areas (e.g. Xi'an city, etc.). WR plays an important role in ecosystem vegetation recovery and household livelihoods.
We also add those explain in Section 2.2 (L.94-97).
Point 4: L.122: Briefly explain what the ArcGIS Zonal Statistics tool does and add a reference.
Response 4:
We explain what the ArcGIS Zonal Statistics tool does and add a reference.
Environmental Systems Research Institute (ESRI), (2014). ArcGIS Desktop Help 10.2 Spatial Analyst. http://resources.arcgis.com/en/help/main/10.2/index.html
Point 5: L.138: “0.542”. Why this value? What is the assumption here? Explain and add supporting reference.
Response 5:
Zhu et al.(2005) calculated the maximum light use efficiency of some typical vegetation types in China. Based on the results of Zhu et al. (2005), the maximum light efficiency of the main vegetation types of Uxin belong to grassland, and it was set to 0.542.
"Please see the attachment."
From: Zhu, W.; Pan, Y.; He, H.; Yang, M.; Long, Z.; Yu, D. In Simulation of maximum radiation conversion efficiency for different vegetation types, IEEE International Geoscience and Remote Sensing Symposium Proceedings, Seoul, Korea, 2005; Seoul, Korea.
Point 6:Table 3: Why these 10°C and 5m/s thresholds? Explain your assumptions!
Response 6:
Our explanation is as follows: accumulated temperature >10 ℃ is the lower limit temperature of vegetation growth, which is widely used in agrometeorological evaluation in China. According to the revised wind erosion equation (RWEQ) model, the wind speed 5 m/s is threshold for beginning of wind erosion.
Point 7: L.221 and further: In which units are you expressing all your slope values? %? Degrees?
Response 7:
Slope is dimensionless. Slope is b value in equation y = a + bx, the higher the b value represents the greater the rate of change.
Point 8: Figure 2: This figure is still hardly readable! Please expand to entire page.
Response 8:
We changed the layout of Figure 2 from landscape to portrait.
Point 9: L.236-241: These are huge numbers – are you sure of them? What do these observations tell you about the local context and ES provision in general? What would be the detrimental implications of not having all this amount of sand trapped by ecosystems?
Response 9:
Because, these huge numbers (SP) were estimated as the difference between potential wind erosion (no vegetation) and actual wind erosion. Similar study are shown inFig.1 ( Ouyang et al., 2016)
Without sand trapped by local ecosystems, the sandstorm disasters would affect people and environment in Uxin and downwind region of Mid-western China. We also add some interpret in L.240.
"Please see the attachment."
Fig. 1 Spatial pattern of ecosystem service provision in China in 2010 (from Supplementary Materials of Ouyang et al., 2016)
Ouyang, Z.; Zheng, H.; Xiao, Y.; Polasky, S.; Liu, J.; Xu, W.; Wang, Q.; Zhang, L.; Xiao, Y.; Rao, E., et al. Improvements in ecosystem services from investments in natural capital. SCIENCE 2016, 352, 1455-1457.
Point 10: L.243-250: A bit strange to express water volumes in tonnes instead of m3. Any explanation? Also, interpret a little bit more. How do these observations differ from the rest of China? What are the implications? Are these numbers good or bad? What is the impact on ecosystems?
Response 10:
Thanks for your reminder. It’s a wrong unit, we changed it to m3.
And we also added a little bit interpret for WR ( L.246 and L.253-256).
Point 11: Figure 3: Expand figure to improve readability.
Response 11:
We changed the layout of Figure 3 to portrait.
Point 12:Table 4: Add a column that briefly describes each variable to help the reader.
Response 12:
We added an table note to briefly describes variables.
L.301-303: The dependent variable ES1–ES2 represents the relationship (correlation coefficient ) between two ES in the 38 years.
Point 13:Section 4.1: I reckon that this section is very descriptive and belongs more to the Results part than the
Discussion. It seems to me that you really start to “discuss” your results in Section 4.2.
Response 13:
Thanks for your comments. According to your suggestion, we move the Section 4.1 to Results part as Section 3.4.

Reviewer 3 Report
The variable types that describe the Driving forces of ecosystem services interactions, are all "internal" to the analyzed system (with the relevant exception of climate change). But would it be interesting to know if there are external variables that influence the driving forces, for example, population change is influenced by the migration balance? Does this in turn affect the demand for ESs? Other variables seem to be important for such a long period: for example, there has been a variation in income / pc (is it likely that) this variation has influenced diets and lifestyles and therefore the demand for ES? it would be equally useful to understand the internal relations between the driving forces: for example the relationship between population variation and LUC, or between LUC and technical progress.
If one of the objectives of the study is to provide tools for the management of the driving forces, it is necessary to remember that even among the policies there are trade-offs and synergies. For an effective management of ESs it is therefore necessary to understand how the driving forces themselves are interrelated.
Author Response
Response to Reviewer 3 Comments
Point 1: The variable types that describe the Driving forces of ecosystem services interactions, are all "internal" to the analyzed system (with the relevant exception of climate change). But would it be interesting to know if there are external variables that influence the driving forces, for example, population change is influenced by the migration balance? Does this in turn affect the demand for ESs?
Response 1:
Thanks for your comments, we think it will be interesting in analysis external variables influence the driving forces. However, the data of our driving forces is mainly from the statistical yearbook, some variable are not easy to obtain. To overcome this shortcoming, follows your comment, we added the discussion about relationship between driving forces , please see Section 4.3 in L.415-426.
Point 2:
Other variables seem to be important for such a long period: for example, there has been a variation in income / pc (is it likely that) this variation has influenced diets and lifestyles and therefore the demand for ES?
Response 2:
Thanks for your comments, the income variation may influence diets and lifestyles of local household. In our another unpublished work, we developed an household survey to understand the linking among demographic factors, land use, ecosystem services. We found that changes in family demographic factors (income, family size, education) enhanced their land use intensity, resulting in an increased supply capacity of ecosystems. We also observed that due to pursuit health perception of local households, vegetables consumption of families was a little higher than meat consumption during last decade. According to our survey, corn and potatoes are mainly planted in the area, vegetables are usually imported from the surrounding provinces(e.g. Shanxi and Shandong).
Owing to lack of long period data in change of income, diets and lifestyles at the regional scale , so we did not carry out this analysis about demand for ES.
Point 3:
It would be equally useful to understand the internal relations between the driving forces: for example the relationship between population variation and LUC, or between LUC and technical progress.
If one of the objectives of the study is to provide tools for the management of the driving forces, it is necessary to remember that even among the policies there are trade-offs and synergies. For an effective management of ESs it is therefore necessary to understand how the driving forces themselves are interrelated.
Response 3:
Thanks for your comment, let us understand the important and useful of relationship between driving factors. Therefore we rewrote the discussion Section 4.3 in L.415-426, As shown below:
According our study, the inner relationship between driving forces should be cautious about. And relationship among drivers may display both trade-offs and synergies or interaction effect. For example, population variation and technical progress would affect the change of land use, and eventually lead to the change of ES supply in this region. Restricted by prohibition of open grazing policy since the early 2000s, herdsmen families in Uxin increased mechanical power in artificial pasture. Such changes in land management increased sheep breeding and reduced grazing pressure on nature grassland [41]. Farm families increased their investment in crop farming and cultivated land during the last three decades, which resulted in cropland expansion hot-spot areas and increase of grain production. It was founded in “Grain for Green” project region in Ningxia, China, the hot-spot areas of 37% of well managed land had contributed 57% of grain output [45]. Similarly, Pretty et al. (2006) found that farming systems adopting sustainability enhancing practices had increased productivity and improving the supply of ES [51].

Round 2
Reviewer 2 Report
Thank you for addressing my comments. I believe that your manuscript has improved in quality through the different versions. Except for the use of English which could still deserve a final proofread by a native speaker, the paper looks now ready for acceptance and publication.
This manuscript is a resubmission of an earlier submission. The following is a list of the peer review reports and author responses from that submission.
Round 1
Reviewer 1 Report
I think that the presentation of the data and results has been improved through the revision of the manuscript. Still, I would like to point to a major flaw in the general approach of the paper.
The key issue is that the authors do not provide a meaningful conceptual definition of ecosystem services and trade-offs. Here is a definition of ecosystem services given by Rodriguez et al. in 2006:
"Ecosystem service trade-offs arise from management choices made by humans, which can change the type, magnitude, and relative mix of services provided by ecosystems. Trade-offs occur when the provision of one ES is reduced as a consequence of increased use of another ES."
Since the authors regress different drivers for each of the ecosystem services, there is no link between the different ES and thus no trade-offs. In my understanding, it makes no sense to describe the trade-off between sheep and sand storm prevention in Figure 3 because there is no common driver i.e., there are not the same explanatory variables in the regression in Table 3. In other words: If the sheep management would completely change, there would be no possible effect on sandstorm prevention because they are explained by different driving factors in time. Thus, there are no trade-offs. It's a correlation that has nothing to do with human choices. If there is an effect e.g. through expanding sheep in space and thus changes in land-use, the presented analysis does not reveal this.
I understand that it can be meaningful to address long-term changes in ecosystem services in space and time. However, I really think that the authors should not frame the analysis in the context of trade-offs and synergies unless they really focus on those ES that are conceptually and analytically linked.
Thus, my suggestion would be that the authors change their title and everything that is associated with trade-offs and keep the analysis or alternatively focus on ES trade-offs, which implies that they need to re-consider their choice of ES.
Some additional comments:
Please do not present additional data in the Discussion section. Figure 4 belongs to the data section.
Please do cite consistently. For example, on line 300 the authors state that sheep/grain and NPP usually exhibit a competitive relationship. I wouldn't expect so and I checked the two references given. Both do not address NPP in any way. Thus, they are inappropriate.
Please unify the y-axes in Figure 3. These are relative measures thus they can be directly compared.
Please use the same colors for the same ES in Figure 2. It's really difficult to interpret the numbers if the color scheme always changes and one can not directly see whether a service increased or decreased.
Reference
Rodríguez, J. P., T. D. Beard, Jr., E. M. Bennett, G. S. Cumming, S. Cork, J. Agard, A. P. Dobson, and G. D. Peterson. 2006. Trade-offs across space, time, and ecosystem services. Ecology and Society 11(1): 28. [online] URL: http://www.ecologyandsociety.org/vol11/iss1/art28/
Reviewer 2 Report
The topic is very interesting, and the case study is significant. The paper is very well organized, and the results are clearly presented and discussed.
Only the following revisions are recommended:
Conclusions.
The § 5 “Conclusions” is only six lines. It must be enlarged inserting some comments about the obtained results in the current research on ecosystem services.
Appendixes
Maybe it is not necessary to insert two appendixes of one page each. The two tables could be simply inserted in the text as two new tables.
Reviewer 3 Report
How do Trade-offs and Synergies between Ecosystem Services Change in the Long Period? The Case Study of Uxin, Inner Mongolia, China
Review
General comments:
An interesting, nicely written and well-presented paper that shows a high level of technicality and provides information about ecosystem services in an understudied region of the world. It shows much of the necessary qualities to be published. However, my main concern is about the very low level of interpretation of the results in the paper. The analysis is conducted thoroughly and shows a high level of attention to details and accuracy, but the paper in general is way to descriptive and not sufficiently interpretative. I would highly recommend the authors to concentrate on improving the narrative. The Discussion and Conclusions need to be improved in that respect. Attention should be placed throughout the paper on stepping away from over-technical discussions and look at the bigger picture of what the results imply. Who is the intended audience? How is this paper useful to them? Provide recommendations based on the observations made through your study in order to advise future land use policies (for instance). The paper needs to be way more convincing in that regard.
Specific comments:
Title:
I would slightly adapt it. Here is a suggestion: “How do trade-offs and synergies between ecosystem services change over time? The case of Uxin, China”
Abstract:
L.21: delete “case”
Introduction:
L37: 60%... worldwide?
L.54: Not sure about the word “unpredictable” here. If these phenomena are totally unpredictable, perhaps there is no need to study them then? I suggest rephrasing it.
L.74: attempted è attempt
Materials and methods:
L.86: subshrubs
Figure 1: Add “A” and “B” to distinguish the two maps. Make the borders of the Uxin region thicker so that they appear more neatly. Add A and B to caption as well. Remove “case” word from caption.
L.93-94: These aren’t “classic” ES categories. Could you specify and explain where these ES indicators fit in classic ES classification schemes like for instance MEA, TEEB or CICES?
L.104-108: This paragraph describing Table 1 should come right before Table 1. (L.100).
L.122-123: Briefly explain what this tool does.
L.138: 0.542. Why this value? What is the assumption here? Add supporting reference.
L.169: performed è conducted
L.186: quality of ES è quality changes in ES
Table 2: I would try to align variables names (top-left corner?) and their description to improve the readability of the table. Also, why these 10°C and 5m/s thresholds? Explain your assumptions.
Results:
L.201: Spatially mean è mean spatial
L.202: h/m2 è what is this unit? Explain.
L.202: highest è higher
L.206 and further: In which units are you expressing all your slope values? X? Degrees?
L.198-207: Please provide more interpretation.
L.209-215: Same comment. What does all this say about the context?
Figure 2: Hardly readable. Please expend to entire page.
L.225-227: NPP are è NPP is
L.229-234: These are impressive numbers – are you sure of them? What do these observations tell you about the local context and ES provision in general? What would be the detrimental implications of not having all this amount of sand trapped by ecosystems?
L.236-243: A bit strange to express water volumes in tonnes instead of m3. Any explanation? Also, interpret a little bit more. How do these observations differ from the rest of China? What are the implications? Are these numbers good or bad? What is the impact on ecosystems?
Figure 3: Expand to entire page to improve readability. “Maps of” in caption should not be in bold. Increase contrast on maps to improve readability.
L.258-264: At first glance, what do these synergies/trade-offs seem to suggest? Provide a couple of sentences of interpretation.
Table 3: Add a column that briefly describes each variable to help the reader.
Discussion:
L.290: period è run
Figure 4: Place it after it is mentioned in the text for the first time.
L.296-297: How is this possible? Explain.
L.297-299: Climate change symptoms tend to go in the opposite direction, i.e. increased occurrence of extreme weather events like cyclones and storms. Explain a little more.
L.301: why usually competitive? Explain.
L.325: Has something similar been observed in the literature?
L.326: Why is “ES” strangely written there?
L.333: decrease trend è decreasing trend
Section 4.2: So, what does that imply for ES provision in general? Interpret. What are the implications for future land use planning policies? What is in there to be learned from your study?
L.348-349: Very strange sentence. What do you mean here?
Section 4.3: Pretty shallow. Acknowledge the limitations of your study and suggest further areas of improvement.
Conclusions:
Way too short. Do not use bullet points. Many readers will only read that section. You should pay more attention to its quality. Make it useful for policy-makers or whoever you intend this study to be dedicated to. Provide interpretations and make recommendations based on your observations.
Appendix B:
Provide 1-2 sentences of explanations to help readers know about the most salient things to understand from these two graphs. The easier you make it for readers, the more they will like and recommend your paper.